# Preparation of Isopropyl Acrylamide Grafted Chitosan and Carbon Bionanocomposites for Adsorption of Lead Ion and Methylene Blue

**DOI:** 10.3390/polym14214485

**Published:** 2022-10-23

**Authors:** Mahmoud Essam Abd El-Aziz, Samir M. M. Morsi, Kholod H. Kamal, Tawfik A. Khattab

**Affiliations:** 1Polymer and Pigments Department, National Research Centre, 33 El Bohoth St., Dokki, Giza P.O. Box 12622, Egypt; 2Water Pollution Research Department, National Research Centre, 33 El Bohouth St., Dokki, Giza P.O. Box 12622, Egypt; 3Dyeing, Printing and Auxiliaries Department, National Research Centre, 33 El Bohoth St., Dokki, Giza P.O. Box 12622, Egypt

**Keywords:** crosslinked chitosan, carbon nanoparticle, nanocomposite, water treatment

## Abstract

Wastewater, which is rich with heavy elements, dyes, and pesticides, represents one of the most important environmental pollutants. Thus, it has been significant to fabricate environmentally friendly polymers with high adsorption ability for those pollutants. Herein, crosslinked chitosan (C-Cs) was prepared using isopropyl acrylamide and methylene bisacrylamide. Carbon nanoparticles (C-NPs) were also obtained by the treatment of the agricultural wastes, which was used with C-Cs to prepare C-Cs/C-NPs nanocomposite (C-Cs/C-NC). Fourier-transform infrared spectroscopy (FTIR), X-ray diffraction (XRD), and transmission electron microscope (TEM) were used to investigate the prepared adsorbent. C-Cs, C-NPs, and C-Cs/C-NC were used in water treatment for the adsorption of lead ions (Pb^+2^) and methylene blue (MB). The adsorption process occurred by the prepared samples was investigated under different conditions, including contact time, as well as different doses and concentrations of adsorbents. The findings exhibited that the adsorption of Pb^+2^ and MB by C-Cs/C-NC was higher than C-Cs and C-NPs. In addition, the kinetic and isotherm models were studied, where the results showed that the adsorption of Pb^+2^ and MB by various adsorbents obeys pseudo-second-order and Langmuir isotherms, respectively.

## 1. Introduction

Water is an essential provenance of life, as it caps about 70% of the territory’s surface. However, only about 0.06% of this water is available as freshwater for drinkable [1]. Despite the massive amount of water on Earth, very small amounts are suitable for consumption [2,3]. Furthermore, human activities, such as industrialization, an energy-intensive lifestyle, urbanization, etc., have contributed to raw-sewage-to-water contamination [4]. Water contamination by heavy metals, pesticides, and organic dyes has received higher attention recently due to their toxicity to the environment, and bioaccumulation, posing a grave menace to both human validity and the ecosystem, where the harmful compounds are infiltrated and concentrated into the food chain, such as fish and other edible organisms throughout their build-up in living tissues [5,6]. Even at low levels, the traces of metals can interfere with the enzymes of living creatures, which are impossible to remove once they have entered the organism, causing negative health consequences [7]. For that, the mean task for the researchers is the purification of wastewater and removal of toxic metals. There are various processing methods that can be used for this intent, such as electrochemical treatment, chemical precipitation, reverse osmoses, membrane technologies, filtration, ion-exchange, adsorption, etc. [8,9]. The adsorption process is the most cost-effective, productive, and simple method [10,11].

Recently, researchers worked to use adsorbents based on natural polymers, such as cellulose, gums, chitosan, sodium alginate, etc., due to their abundance, non-toxicity, biodegradability, cheap cost, etc. [12,13,14]. They were usually used in the flocculation process or as adsorbents [15,16]. Chitosan has been deemed as one of the important biopolymers, which is a derivative of chitin obtained from crustacean shells [17]. It is inexpensive and is one of the most abundant biopolymers and has gotten high attention in wastewater treatment, owing to its numerous amino and hydroxyl groups. However, there are some limitations for practical application due to the weakened mechanical strength and solubility in acidic environments, plus weak adsorption capacity, besides the shortage of selectivity [18].

At the present time, nanotechnology is being used in many applications, such as agriculture, drug delivery, water purification, etc. [19,20,21]. Nanotechnology is the phenomenon of the application of materials on a nanometer-scale level that is usually measured in the range of 1 to 100 nm. Nanomaterials can be prepared in various forms, such as nano-wires, sheets, tubes, particles, and quantum dots. In wastewater treatment, many nanomaterials can be applied due to their efficiency, eco-friendliness, and unrivaled functionalities for the decontamination of wastewater [22,23]. Nanomaterials are a class of adsorbents that have received growing attention recently. They have the ability to adsorb both inorganic and organic compounds from aqueous solutions. In addition, they have the ability to kill and remove microorganisms such as *Pseudomonas Aeruginosa*, *Escherichia Coli*, *Candida Albicans*, and *Staphylococcus Aureus* from water [24,25].

Herein, crosslinked chitosan (C-Cs) was prepared by the grafting of N-isopropyl acrylamide onto the chitosan skeleton and using N,N′-methylene bisacrylamide as a crosslinker. In addition, agricultural wastes obtained from trimming trees, a rich source of carbon, were used in the preparation of carbon nanoparticles. Accordingly, a new nanocomposite adsorbent was prepared by incorporation of C-Cs to crosslinked chitosan during grafting polymerization of chitosan to apply as adsorbent for lead ions and methylene blue.

## 2. Methodology

### 2.1. Materials

Lead acetate, nitric acid (HNO_3_), hydrochloric acid (HCl), and sodium hydroxide (NaOH) were obtained from Al-Gomhoria Co., Cairo, Egypt. Chitosan (medium molecular weight, deacetylation >75%), N-isopropyl acrylamide, N, N′-methylene bisacrylamide and methylene blue (MB) were obtained from Sigma. All solvents and reagents (analytical grade) were applied without additional purification, and all aqueous media were prepared using distilled water.

### 2.2. Preparation of Carbon Nanoparticles (C-NPs)

The agricultural wastes obtained from the trim tree were treated by pyrolysis at 300 °C in the absence of air to obtain the biochar [26,27]. The latter was mill ground and sieved using Mesh Testing Sieve No. 400 to obtain a very fine powder of biochar. The biochar powder was treated with nitric acid (1 N) to obtain active sites, and then washed with distilled water till a neutral pH was reached. The collected powder was dispersed in water and then left to settle down where the upper part containing floating carbon nanoparticles (C-NPs) was taken. The latter step was repeated three times to guarantee that the solution contained only C-NPs [28]. Finally, the floating solution containing C-NPs was subjected to centrifugation for 20 min at 10,000 rpm, and then the wet powder was exposed to drying at 60 °C for 24 h to obtain C-NPs.

### 2.3. Preparation of Cross-Linked Chitosan/C-NPs Nanocomposite

Cross-linked chitosan (C-Cs) was prepared by dissolving about 2 g chitosan (Cs) in an aqueous solution of acetic acid (1%). Then 2 g of N-isopropyl acrylamide and 0.02 g N,N′-methylene bisacrylamide were added to the reaction solution under stirring at 300 rpm. A potassium persulfate solution was added to the reaction mixture and the temperature was raised to 70 °C to initiate the crosslinking polymerization process. However, for the preparation of cross-linked chitosan/C-NPs (C-Cs/C-NC), the same previous experiment was carried out, but in the presence of 0.02 g dispersed C-NPs. The reaction solution was dried using a freeze dryer to obtain C-Cs/C-NC.

### 2.4. Instrumentation

FTIR analysis (400–4000 cm^−1^) was carried out for all prepared samples using the KBr procedure on a Mattson 5000 spectrometer (Unicam, Cambridgeshire, UK).

The morphological structure of C-NPS was studied by TEM (JEM-1230, JEOL Ltd., Tokyo, Japan), where a drop of a dispersed C-NPS in water was placed on a copper grid coated by carbon and then insert into a TEM device after drying in the air.

The morphological properties of the prepared samples were performed by SEM, (Quanta-250, Tokyo, Japan). The sample was placed on the plate and exposed to coating with gold using an EMITECH K550X sputter coater, England for 1 min.

The crystal structure of the prepared samples was obtained by an X-ray diffractometer (XRD; Diano, WO, USA), where it used CoKα as a source of radiation which energized at 45 kV. XRD patterns were recorded by CuK radiation supply (λ = 0.154 nm).

The BELSORP MINI-X apparatus (Japan) was used in the study of the adsorption-desorption isotherm of N2 for the prepared samples at 77 K to measure the surface area and pore size using the Brumauer–Emmett–Teller method (BET).

### 2.5. Adsorption Measurements

#### 2.5.1. Effect of Adsorbent Content

The removal efficiency of the adsorbent doses by adding (0.125, 0.25, 0.5, and 1 g) of C-Cs, C-NPs, and C-Cs/C-NC for removal of 100 mg/L Pb^+2^ and 10 mg/L MB were evaluated. The solutions were filtered, and the concentration of Pb^+2^ and MB was determined by atomic absorption and UV-spectrophotometer, respectively. The percentage of removal efficiency (R%) of adsorbents was estimated by Equation (1).
(1)$$R=\left(\frac{C_{o}-C_{t}}{C_{o}}\right)*100$$
where *C_t_* is the total content of the remaining contaminants after time (*t*; mg L^−1^), and *C_o_* is the initial contaminants’ content (mg L^−1^) of ions. 

#### 2.5.2. Effect of Contact Time

The removal efficiency of Pb^+2^ (100 mg/L) and MB (10 mg/L) by the C-Cs, C-NPs, and C-Cs/C-NC was studied at numerous different times (5–120 min), pH = 6 using 1 g/L of adsorbent. The removal efficiency percent (R%) was determined by Equation (1). The quantity of Pb ions or MB uptake by 1 g adsorbent (*q*) was estimated by Equation (2).
(2)$$q=\left(C_{o}-C_{t}\right)x\;\frac{V}{M}$$
where *C_o_* is the total content of the initial contaminants (mg/L), *C_t_* is the total content of the remaining contaminant after a time (*t*) (mg/L), *M* is the mass of the adsorbent added (g), and *V* is the solution volume (mL).

#### 2.5.3. Effect of the Contaminants (Pb Ions and MB) Concentrations on Removal Efficiency and Adsorption Capacity

The R% and q were calculated for the prepared adsorbents at various concentrations of Pb^+2^ (10, 30, 50, 100, and 150 mg/L) and MB (1, 3, 5, 10, and 15 mg/L) were investigated at the time 120 min and pH 6 using 1 g/L of the adsorbent.

#### 2.5.4. Kinetic Measurements

In order to study the adsorption kinetics of Pb^+2^ and MB using C-Cs, C-NPs, and C-Cs/C-NC as adsorbents, four kinetic models were applied, which are pseudo-first-order, pseudo-second-order, intraparticle diffusion model, and Elovich.
I.Pseudo-first-order kinetics are denoted by
(3)$$\log\left(q_{e}-q_{t}\right)=\log q_{e}-\frac{k_{1}}{2.303}\;t$$
where *q_t_* is the uptake capacity (mg/g) at any given time (*t*, min), *q_e_* is the equilibrium uptake capacity (mg/g), and *k*_1_ is the pseudo-first-order rate constant (min^−1^) which can be estimated from the slope of the logarithm line graph (*q_e_* − *q_t_*) against t [29].II.Pseudo-second-order kinetics were applied to verify the constant adsorption rate according to the subsequent equation:(4)$$\frac{t}{q_{t}}=\frac{1}{K_{2}q_{e}}+\left(\frac{1}{q_{e}}\right)t$$
where *K*_2_ represents pseudo-second-order rate constant (g/mg min) which can be analyzed from the intercept of the linear plot of *t*/*q_t_* versus *t*. The following expression indicates the rate of adsorption *h* (mg/g min) [30]:(5)$$h=k_{2}q_{e}^{2}$$III.The intra-particle diffusion kinetic model can be denoted as
(6)$$q_{t}=k_{p}{\left(t\right)}^{0.5}+c$$
where *k_p_* represents the intraparticle diffusion rate (mg.min^1/2^/g), and C represents the constant which can be expressed from the slope and intercept, respectively, of the line graph of *q_t_* vs. *t*^0.5^ [31].IV.The Elovich kinetic model can be described by
(7)$$q_{t}=\frac{1}{\beta }ln\left(\alpha \beta \right)+\frac{1}{\beta }ln\left(t\right)$$
where *β* is the constant of desorption (mg.min/g), and α is the initial adsorption rate (mg/g.min), which can be determined from the slope and intercept, respectively, for the linear plot of *q_t_* versus *ln t* [32].

#### 2.5.5. Isotherm Study

Adsorption isotherms demonstrate how the adsorbent particles interact with the adsorbate particles. The interaction of Pb^+2^ and MB with C-Cs, C-NPs, and C-Cs/C-NC as adsorbents were fixed by using four isotherm models.

I.Langmuir isotherm assumes monolayer adsorption which is epitomized by the subsequent equation
(8)$$\frac{q_{t}}{q_{max}}=bC_{t}/\left(1+bC_{t}\right)$$
where *q_t_* represents the uptake capacity after time *t* (mg/g), *C_t_* represents the concentration after time *t* (mg/L), *q_max_* is the maximum uptake capacity (mg/g), and *b* represents Langmuir constant (L/mg), which is associated with the adsorption energy.

The separation factor (*R_L_*) is the basic property of the Langmuir isotherm which is described as
(9)$$R_{L}=1/\left(1+bC_{o}\right)$$

*R_L_* > 1 signifies unfavorable adsorption, *R_L_* = 1 signifies linear adsorption, and *R_L_* = 0 translates into irreversible, whereas *R_L_* values between 0 and 1 specify promising adsorption [33].

II.The Freundlich isotherm model, which assumes a multilayers adsorption, is denoted by the following equation:(10)$$lnq_{e}=lnk_{f}+\left(\frac{1}{n}\right)lnC_{e}$$
where *k_f_* is the constant of Freundlich, and *n* is the adsorption strength which can be expressed from the intercept and slope, respectively, of the linear graph of *ln q_e_* versus *ln C_e_*. *n* = 1 signifies linear adsorption; *n* < 1 designate chemical process; and *n* > 1 represents the physical process [34].III.The Temkin isotherm model suggests that the sorption energy during the process of adsorption decreases linearly as a function of increasing adsorption site saturation. The Temkin isotherm can be given by Equation (11) (11)$$q_{e}=B\;ln\;k_{t}+B\;ln\;C_{e}$$
where *K_t_* is the equilibrium binding constant (mol/L), *B* is the constant associated with the adsorption heat *B = RT/b*, *b* is the Temkin constant related to the adsorption energy, *R* is the gas constant (0.00813 kJ/mol^−1^), and *T* is the temperature (K).

If the constant *B* is less than 8 kJ/mol, it specifies a weak interaction between the adsorbate and adsorbent, so such adsorption is considered physical adsorption [34].

IV.The Dubinin–Radushkevich (D–R) isotherm can be given in the following form [35]:(12)$$ln\;q_{e}=ln\;q_{m}-\beta {Ɛ}^{2}$$
where *β* is the activity coefficient, and *Ɛ* is Polanyi potential, which is defined as
(13)$$Ɛ=RT\;ln\left(1+\frac{1}{C_{e}}\right)$$

Sorption energy (*E*) is the energy used (kJ/mol) when the Pb^+2^ and MB transferred to the surface of the C-Cs, C-NPs, and C-Cs/C-NC, which can be described as
(14)$$E=\frac{1}{{\left(2\beta \right)}^{0.5}}$$

## 3. Results and Discussion

### 3.1. Morphological Studies

Figure 1 illustrates the transmission electron microscope micrographs and scanning electron microscope of the prepared C-NPs. Figure 1a illustrates the spherical shapes of the prepared C-NPs, which have a narrow size distribution and the majority of particle sizes are less than 100 nm. Furthermore, the SEM image (Figure 1b) shows homogeneous and spherical particles with narrow particle size distribution.

The surface morphology of C-Cs and C-Cs/C-NC is illustrated in Figure 2a,b, respectively. Generally, the images clearly point out the difference in surface morphology between the cross-linked chitosan and the nanocomposite. The micrograph of C-Cs shows a relatively smooth surface interspersed with some pores of small size. The crosslinked polymer appears as massive matrices with dense and smooth hemispherical and irregular structures on the surface. However, the C-NPs in C-Cs/C-NC contribute to changing the morphology into a rough surface filled with highs and lows. The image shows denser pores than C-Cs image with the emergence of some huge caves.

The X-ray diffraction patterns of C-Cs, C-NPs, and C-Cs/C-NC are recorded in Figure 3. The XRD pattern of C-NPs shows the crystal plane index (0 0 2) at 2θ = 24.85°. The emergence of this broad peak indicates the existence of stacking amorphous carbon structures with random parallel and horizontal orientation of aromatic sheets. Another diffraction peak appeared at 2θ = 43.65° indicating that the carbon crystals are stacked in an ordered turbostratic structure [36,37,38].

The X-ray diffraction of C-Cs shows the two distinct peaks of chitosan at 2θ = 8.19° and 20.24°. These two peaks indicate the ordered crystal structure created by hydrogen bonds in chitosan structure. In addition, other peaks at 2θ = 29.74°, 30.94°, 37.07°, 40.86°, 43.22° and 48.83° indicating the formation of crosslinked chitosan nanoparticles [39]. The XRD pattern of C-Cs/C-NC shows a high match with that of C-Cs where the characteristic broad peaks of chitosan appeared at 2θ = 12.11° and 21.35°. The other sharp peaks also emerge at 2θ = 29.77°, 30.89°, 37.05°, and 40.37°. The presence of C-NPs in C-Cs/C-NC can be proved by the peak indexed as 002 that emanates as a shoulder at 2θ = 24.16°, and the other peak at 2θ = 43.55°.

FT-IR spectra of C-Cs, C-NPs, as well as the synthesized C-Cs/C-NC were illustrated in Figure 4. The crosslinked Cs have a broad peak at 3300 cm^−1^ conformable to O-H and N-H stretching vibration, and the basic characteristic bands at 1650, 1550 and 1050 cm^−1^ are congruous to C=O, the N-H stretching of a primary amine group vibration, and C-O-C stretching vibration, respectively [40]. In addition, the carbon nanoparticles showed characteristic bands of the common peaks at 3400, 1650 and 1050 cm, owing to the stretching vibrations of O-H, C=O, and C-O, respectively [41]. These peaks indicate the high functionality of the prepared carbon nanoparticles, which qualifies them to use as sorbents. The prepared nanocomposite (C-Cs/C-NC) contains distinct groups for both C-Cs and C-NPs as illustrated in Figure 4.

The surface and pore lineaments of the prepared crosslinked chitosan and their nanocomposite with carbon nanoparticles were examined using the BET method, and their values are summarized in Table 1. Evidently, there is a considerable change in the surface area of crosslinked chitosan by incorporation of carbon nanoparticles, where the surface area raised from 24.5 for C-Cs to 30.3 m^2^/g for C-Cs/C-NC. This is the cause for the higher adsorption of lead ions and MB by C-Cs/C-NC over C-Cs and C-NPs. In addition, the average size of pores is less than 50 nm, which indicates the mesoporous structure of the prepared adsorbents [42,43].

As maintained by the classification of Brunauer–Deming–Deming–Teller [44], the adsorption–desorption isotherms for all adsorbents samples exhibit type III with hysteresis loop of type H3 as shown in Figure 5, which is distinguishing for the mesoporous structure of the prepared adsorbents [45,46].

### 3.2. Adsorption of Pb^+2^ and MB

The adsorbent amount has a significant value in the adsorption process, and it defines the adsorbent capacity through the number of active binding sites available to remove the pollutants from solutions. Figure 6a,d illustrates the effect of changing the dose of C-Cs, C-NPs and C-Cs/C-NC from 0.125 to 1 g on the uptake capacity of lead ions and MB, respectively, at pH 6, time 120 min, and 100 mg/L of adsorbent concentration. The findings display that the adsorption is increased as the adsorbent dose from 0.125 to 1 g, which may be due to the increase of active site numbers that are available to adsorb ions [47]. In addition, the order of adsorption of Pb ions and MB by various adsorbents was C-Cs/C-NC > C-NPs > C-Cs.

An additional increase in the percentage of the removal of metal ions of adsorbent dosage was detected when the same doses were used after adding the activated carbon nanoparticles. The results exhibited that the removal percentage of lead ions at a dosage of 0.125 g of C-Cs/C-NC (Figure 6a) was 33.6% which increased to 72.8% by increasing the dosage to 1 g, while for MB, the R% was increased from 25.6% to be 80.1% by increasing the dosage of adsorbent to 1 g (Figure 6d).

Figure 6b,e represents the effect of 1 g of C-Cs, C-NPs and C-Cs/C-NC on the removal efficiency of Pb^+2^ and MB, respectively, by varying the contact time from 5 min to 120 min at pH = 6. The removal efficiency was found to increase with growing the contact time, where it increased sharply during the first 60 min then slowly from 60 to 120 min until reaching equilibrium. The fast initial adsorption rate could be attributed to the availability of sufficient vacancies and the high driving force of the ions within the adsorbents, which causes the rapid transfer mechanism between lead ions and MB and the binding site of the C-Cs, C-NPs and C-Cs/C-NC [48]. The slow removal performance is due to active sites becoming exhausted [9].

The results showed that the removal efficiency of Pb^+2^ and MB using C-Cs/C-NC was 72.8 and 80.1, respectively. This may be due to the addition of carbon nanoparticles to crosslinked chitosan improving the removal percentage by increasing the active sites that are capable of absorbing ions and the numbers as well as the pore size of the prepared nanocomposites.

Figure 6c,f represents the effect of 1 g of C-Cs, C-NPs and C-Cs/C-NC on the R% of various concentrations of lead ions and MB, respectively, after 120 min and at pH = 6. The data show that the R% decreased by increasing the concentration of lead ions and MB. This could be attributed to the rarer number of active sites, as well as the surface area of different adsorbents, was constant against the increase of the ions in the solution. So, at the lower ions concentration in the solution, the loading capacity of ions in the adsorbent was high, and so the residual ions concentration in the solution was decreased [49].

### 3.3. Kinetic and Isothermal Studies

The kinetic models, including pseudo-first-order, pseudo-second-order, Elovich and intra-particle diffusion models, as well as the adsorption isotherms, including Freundlich, Langmuir, Temkin and Dubinin–Radushkevich (D–R), were employed to study the kinetics and interaction between lead ions and MB with adsorbents. The kinetic and isotherms model’s constants and correlation coefficients of C-Cs, C-NPs, and C-Cs/C-NC for adsorption of Pb^+2^ and MB were determined and stated in Table 2 and Table 3, respectively.

From the findings summarized in Table 2, due to the correlation coefficient super value (R^2^) of the pseudo-second-order model, and the close reliability among the experimental and uptake capacities [50], we concluded that the adsorption study of Pb^+2^ and MB obeys the pseudo-second-order mechanism (Figure 7 and Figure 8, respectively). Many kinds of literature illustrated that the adsorption kinetics studied of divalent metals follow the pseudo-second-order mechanism [51].

The equilibrium between the lead ions and MB removal and the adsorbent’s surface was studied using various isotherm models to determine the adsorption isotherm (Figure 7 and Figure 8, respectively). The data obtained from equilibrium isotherm using various isotherm models were estimated and are summarized in Table 3; they give important knowledge about the mechanisms of adsorption as well as the adsorbent surface properties and the relationship between solution and the adsorbent.

The results demonstrate that the adsorption of Pb^+2^ and MB provided the best fit with the Langmuir model, which is attributed to the higher value of the correlation coefficient (R^2^). This proposes that the fixation of lead ions and MB is performed in a monolayer, and on energetically equivalent sites (homogenous sites) without interaction between the adsorbed molecules [52]. Remarkably, from the data, the value of n > 1 in Freundlich, B < 8 in Temkin and, E < 8 in (D-R) model validates that the adsorption is weak interaction (physical process). Furthermore, the separation factor (RL) is monitored between 0 and 1, which mentions promising adsorption between sorbates and adsorbents [47].

## 4. Conclusions

In this study, carbon nanoparticles as well as crosslinked chitosan-*N*-isopropylacrylamide were prepared from agricultural residues and *N*,*N*′-methylenebisacrylamide, respectively. A nanocomposite of crosslinked chitosan/carbon nanoparticles was also fabricated. The prepared materials were used as sorbents for lead ions and methylene blue from their solutions. The carbonized structure and nanoscale of the prepared carbon particles were confirmed by XRD pattern and TEM image, which showed average sizes less than 100 nm. Adsorption studies have shown that the greater the amount of sorbents in the solution from 0.125 g to 1 g, as well as the soaking time of these sorbents, the higher the removal of lead ions and methylene blue from the solutions. Additionally, the adsorption efficiency of crosslinked chitosan/carbon nanocomposite, at all dosage concentration, was higher than that of both carbon nanoparticles and crosslinked chitosan. BET analysis showed that the surface area of the nanocomposite (30.35 m^2^/g) is higher than both carbon nanoparticles (14.82 m^2^/g) and crosslinked chitosan (24.56 m^2^/g). This explains the higher adsorption efficiency of the nanocomposite than carbon nanoparticles and crosslinked chitosan. The kinetic adsorption study of Pb^+2^ and methylene blue obeyed the pseudo-second-order mechanism. However, their isotherm studies provided the best fit with the Langmuir model. 

## Figures and Tables

**Figure 1 polymers-14-04485-f001:**
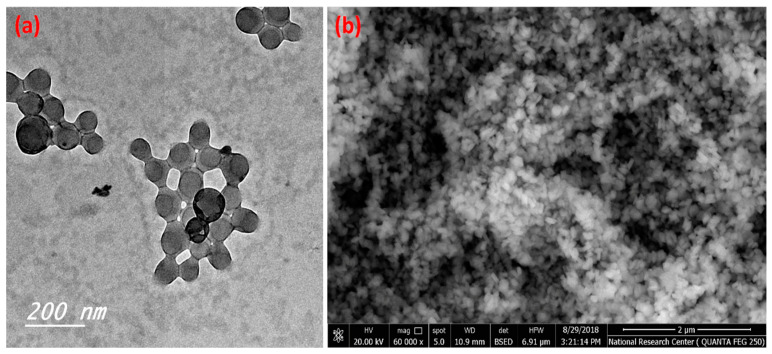
TEM (**a**) and SEM (**b**) of the prepared C-NPs.

**Figure 2 polymers-14-04485-f002:**
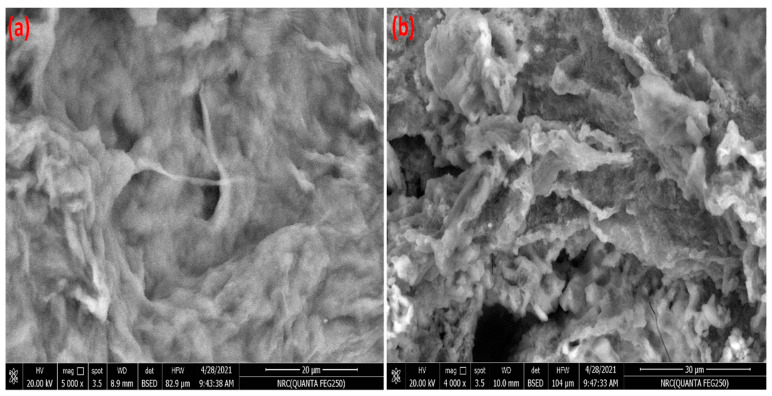
SEM image of C-Cs (**a**) and C-Cs/C-NC (**b**).

**Figure 3 polymers-14-04485-f003:**
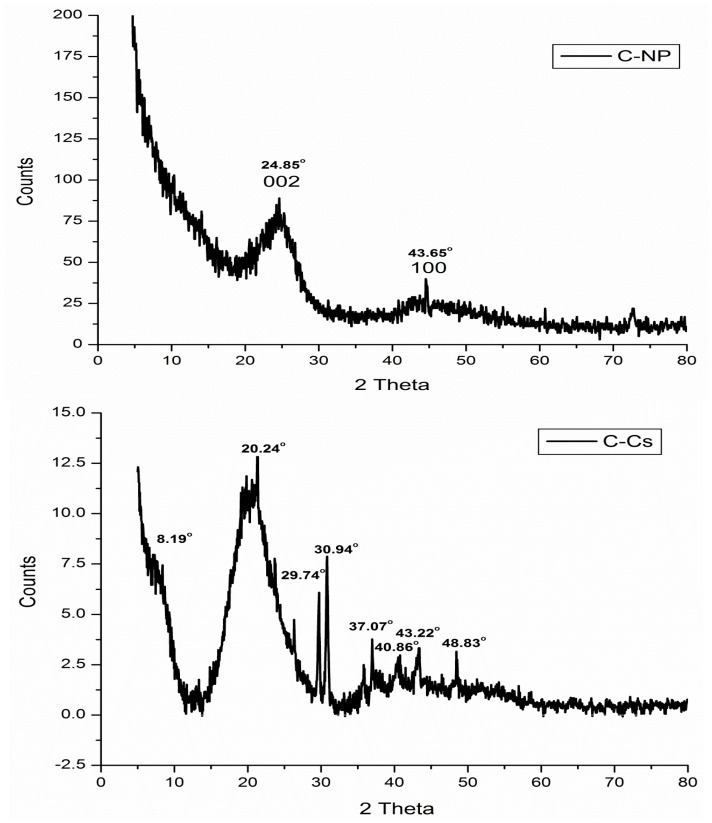
XRD patterns of C-NPs, C-Cs, and C-Cs/C-NC.

**Figure 4 polymers-14-04485-f004:**
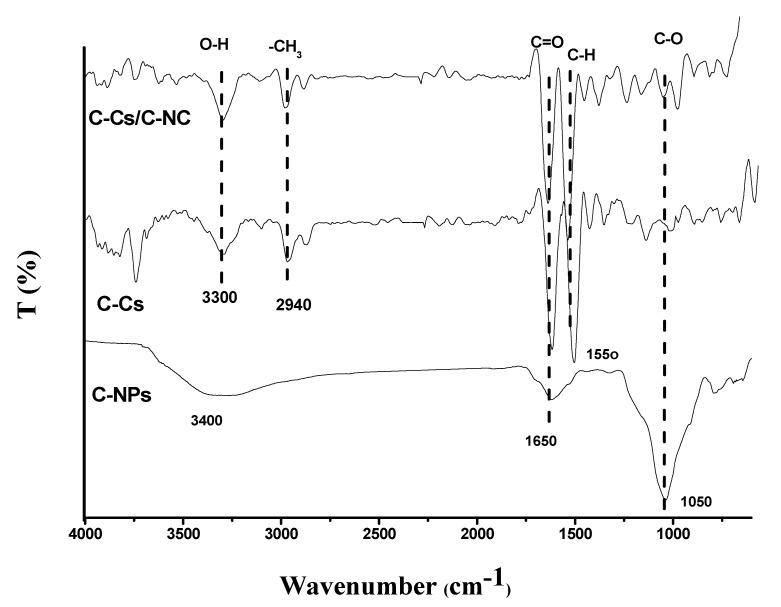
FTIR of C-Cs, C-NPs, and C-Cs/C-NC.

**Figure 5 polymers-14-04485-f005:**
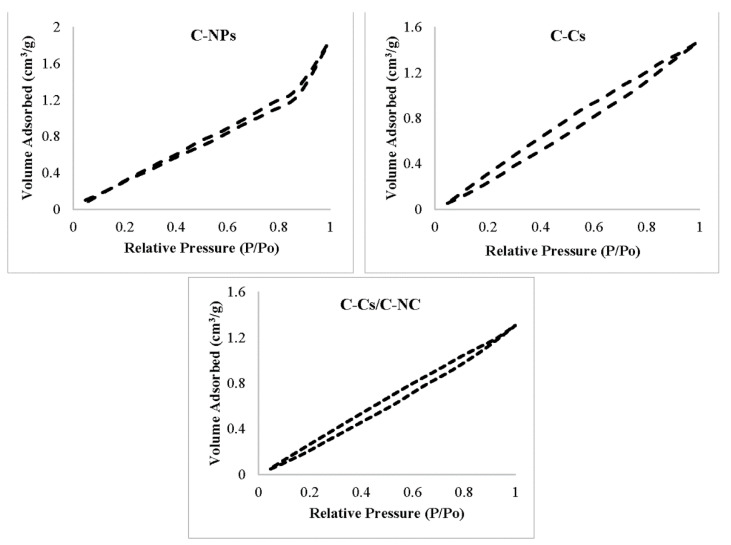
Adsorption–desorption isotherm models of N_2_ at 77 K on prepared samples.

**Figure 6 polymers-14-04485-f006:**
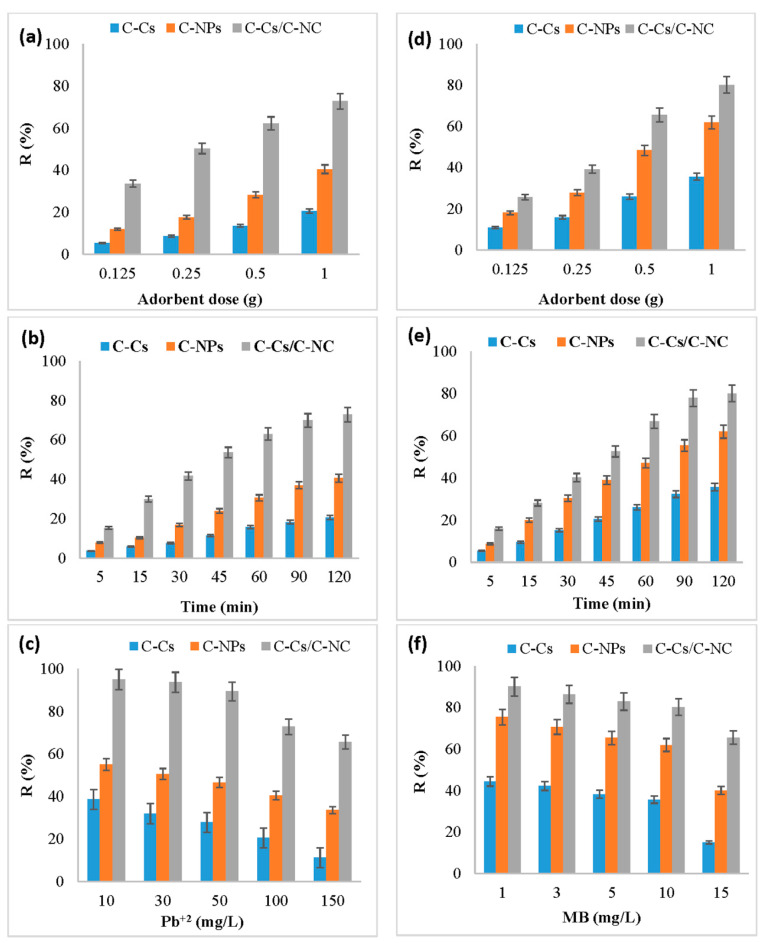
Effect of concentration, contact time, and adsorbent dose on removal percentage for Pb^+2^ (**a**–**c**, respectively) as well as MB (**d**–**f**, respectively).

**Figure 7 polymers-14-04485-f007:**
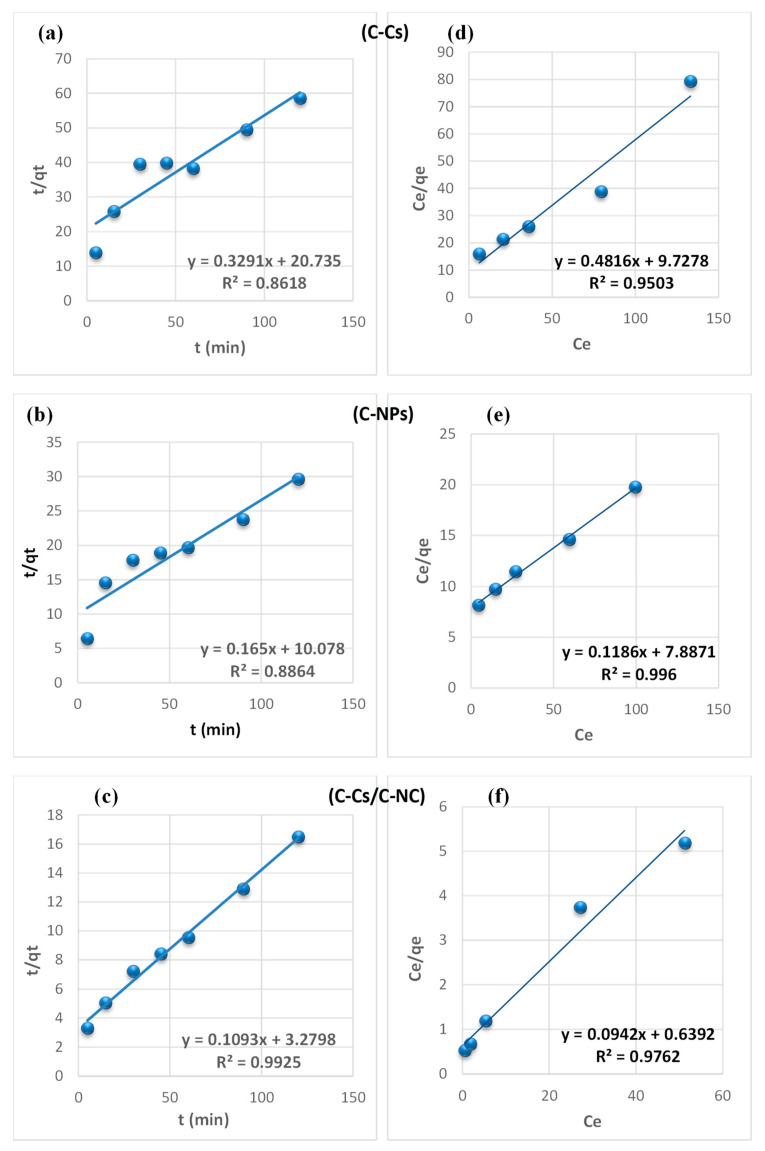
Pseudo-second-order fitting model (**a**–**c**) and Langmuir isothermal fitting model (**d**–**f**) of the removal of lead ions using C-Cs, C-NPs, and C-Cs/C-NC, respectively.

**Figure 8 polymers-14-04485-f008:**
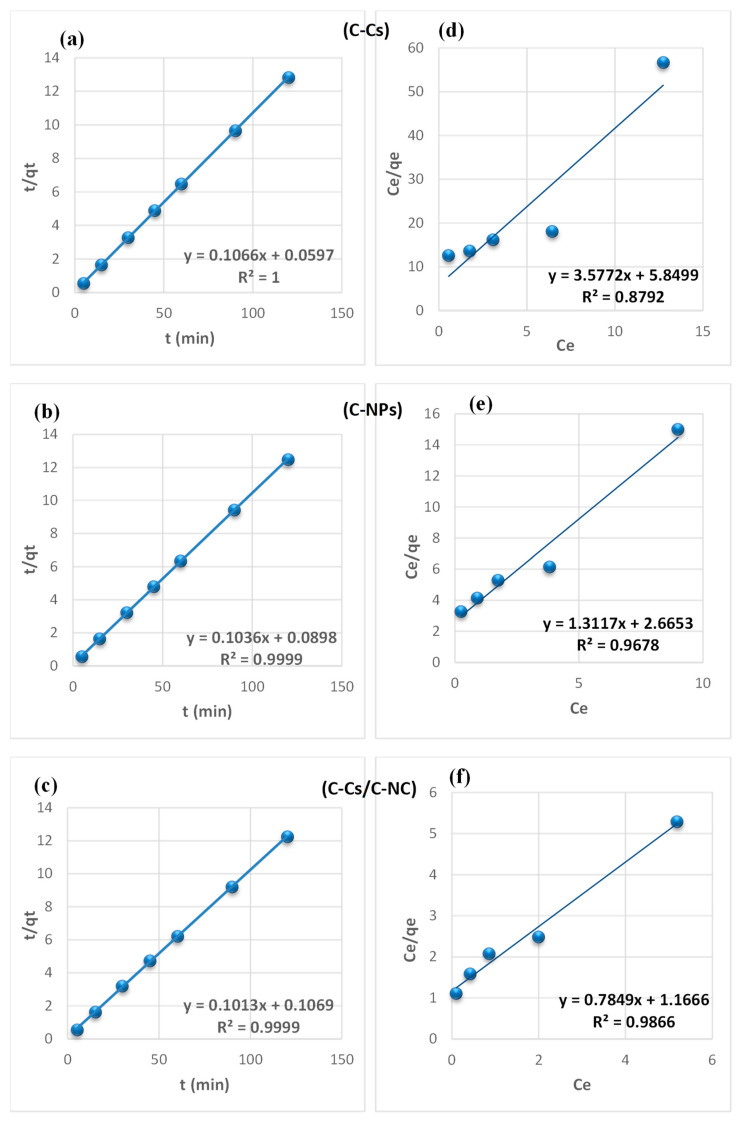
Pseudo-second-order fitting model (**a**–**c**) and Langmuir isothermal fitting model (**d**–**f**) of the removal of MB using C-Cs, C-NPs, and C-Cs/C-NC, respectively.

**Table 1 polymers-14-04485-t001:** The surface characteristics of the prepared C-Cs and C-Cs/C-NC.

Sample	Average Pore Radius (nm)	Surface Area (m²/g)	Pore Volume (cm^3^/g)	Total Pore Volume (cm^3^/g)
**C-NPs**	0.9	14.8	0.0189	0.0117
**C-Cs**	1.6	24.6	0.0172	0.0192
**C-Cs/C-NC**	1.8	30.3	0.0238	0.0267

**Table 2 polymers-14-04485-t002:** Parameters of kinetics for Pb^2+^ and MB removal onto C-Cs, C-NPs, and C-Cs/C-NC.

	Pseudo-First-Order							
	*K_1_* (min^−1^)		*q_e_* (exp.) (mg/g)		*q_e_* (cal.) (mg/g)		*R^2^*	
	Pb^2+^	MB	Pb^2+^	MB	Pb^2+^	MB	Pb^2+^	MB
**C-Cs**	0.04	0.02	2.05	9.35	4.2	0.40	0.8296	0.9632
**C-NPs**	0.05	0.02	4.05	9.60	9.7	0.63	0.8494	0.9939
**C-Cs/C-NC**	0.05	0.04	7.28	9.80	12.9	1.10	0.9094	0.9345
	**Pseudo-Second-Order**							
	***K_2_* (g/mg min)**		***q_e_* (exp.) (mg/g)**		***q_e_* (cal.) (mg/g)**		** *R^2^* **	
	**Pb^2+^**	**MB**	**Pb^2+^**	**MB**	**Pb^2+^**	**MB**	**Pb^2+^**	**MB**
**C-Cs**	0.005	0.19	2.05	9.35	3.0	9.38	0.8618	1.0
**C-NPs**	0.003	0.12	4.05	9.60	6.1	9.65	0.8864	0.999
**C-Cs/C-NC**	0.003	0.09	7.28	9.80	9.1	9.87	0.9925	0.999
	**Intra-Particle Diffusion Model**							
	***K_p_* (mg. g^−1^min^1/2^)**				** *C* **		** *R^2^* **	
	**Pb^2+^**		**MB**		**Pb^2+^**	**MB**	**Pb^2+^**	**MB**
**C-Cs**	0.2		0.03		0.2	8.96	0.9706	0.9879
**C-NPs**	0.4		0.06		0.38	8.96	0.9752	0.9928
**C-Cs/C-NC**	0.69		0.08		0.36	8.98	0.9627	0.974
	**Elovich Model**							
	***β* (mg. min/g)**				***α* (mg/g.min)**		** *R^2^* **	
	**Pb^2+^**		**MB**		**Pb^2+^**	**MB**	**Pb^2+^**	**MB**
**C-Cs**	1.8		10.3		0.13	2.84819 × 10^38^	0.886	0.9136
**C-NPs**	0.9		5.86		0.27	3.55255 × 10^21^	0.8945	0.9669
**C-Cs/C-NC**	0.52		4.56		0.7	4.73069 × 10^16^	0.9784	0.9446

**Table 3 polymers-14-04485-t003:** Parameters of isotherms for Pb^2+^ and MB removal onto C-Cs, C-NPs, and C-Cs/C-NC.

	Langmuir Isotherm							
	*q_max_* (mg/g)			*b* (L/mg)			*R^2^*	
	Pb^2+^	MB		Pb^2+^		MB	Pb^2+^	MB
**C-Cs**	2.07	2.79		0.049		0.06	0.9503	0.8792
**C-NPs**	8.40	7.60		0.015		0.05	0.9960	0.9678
**C-Cs/C-NC**	10.60	12.7		0.150		0.06	0.9762	0.9866
	**Freundlich Isotherm**							
	** *n* **			** *K_f_* **			** *R^2^* **	
	**Pb^2+^**	**MB**		**Pb^2+^**		**MB**	**Pb^2+^**	**MB**
**C-Cs**	1.90	1.7		0.18		0.08	0.8872	0.8004
**C-NPs**	1.38	1.6		0.20		0.20	0.9990	0.9328
**C-Cs/C-NC**	2.11	0.4		1.67		1.59	0.9539	0.9729
	**Temkin Isotherm**							
	** *k_t_* ** **(mol/L)**			** *B* **			** *R^2^* **	
	**Pb^2+^**	**MB**		**Pb^2+^**		**MB**	**Pb^2+^**	**MB**
**C-Cs**	0.38	3.66		0.50		0.08	0.8733	0.6668
**C-NPs**	0.24	5.50		1.48		0.16	0.9535	0.9101
**C-Cs/C-NC**	2.66	2.70		0.85		0.55	0.9716	0.9410
	**(D–R) Isotherm**							
	***qmax* (mg/g)**		** *B* **		***E* (kJ/mol)**		** *R^2^* **	
	**Pb^2+^**	**MB**	**Pb^2+^**	**MB**	**Pb^2+^**	**MB**	**Pb^2+^**	**MB**
**C-Cs**	1.55	0.24	1 × 10^−5^	3 × 10^−7^	0.1	0.58	0.8789	0.8852
**C-NPs**	3.17	0.48	7 × 10^−6^	1 × 10^−7^	0.12	1	0.7917	0.8835
**C-Cs/C-NC**	6.10	0.69	3 × 10^−7^	6 × 10^−4^	0.58	1.29	0.8284	0.8807

## Data Availability

All relevant data are within the manuscript and available from the corresponding author upon request.

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
