# Peer review of "Preparation of Isopropyl Acrylamide Grafted Chitosan and Carbon Bionanocomposites for Adsorption of Lead Ion and Methylene Blue"

_polymers, 2022, doi:10.3390/polym14214485_

Round 1

Reviewer 1 Report

The work based on bio-nanocomposite is promising. However, the are some revisions need to be done before acecptance.

1) What is the novelty in this chitosan-based nanocomposite?

2) From Table 2, the surface area can be rounded and reported in term of 14.8 not 14.82, and so on.

3) The quality of Fig. 3 can be improved by re-plotting for better understanding of the type of adsorption isotherm from IUPAC.

4) The models of adsorption (apart from Langmuir and Freundlich models) are need to be included in the revised manuscript.

5) Table 1 should be removed.

6) Fitting of all adsorption model should be added.

7) How can you explain the sustainable removal of pollutant by adsorption. In general, people say that adsorption s just moving the pollutant from solution to the solid phase resulting in creation of secondary pollutant.

Author Response

Reviewer # 1:

The work based on bio-nanocomposite is promising. However, the are some revisions need to be done before acecptance.

  • What is the novelty in this chitosan-based nanocomposite?

Response: A crosslinked chitosan (C-Cs) was prepared by the grafting of N-isopropyl acrylamide onto the chitosan skeleton and using N,N'-methylene bisacrylamide as a crosslinker. In addition, agricultural wastes obtained from trimming trees, a rich source of carbon, were used in the preparation of carbon nanoparticles. Accordingly, a new nanocomposite adsorbent was prepared by incorporation of C-Cs to crosslinked chitosan during grafting polymerization of chitosan to apply as adsorbent for lead ions and methylene blue.

  • From Table 2, the surface area can be rounded and reported in term of 14.8 not 14.82, and so on.

Response: Table 2 was changed to be Table 1, where all values of surface area were approximated.

  • The quality of Fig. 3 can be improved by re-plotting for better understanding of the type of adsorption isotherm from IUPAC.

Response: Fig. 3 was changed to Fig. 5 and its quality of was improved.

  • The models of adsorption (apart from Langmuir and Freundlich models) are need to be included in the revised manuscript.

Response: All the models of kinetics study (Pseudo-first-order, Pseudo-second-order, Intra-particle diffusion and Elovich) as well as the models of isotherm study (Langmuir, Freundlich, Temkin and Dubinin–Radushkevich) were studied.

  • Table 1 should be removed.

Response: Table 1 was deleted, and the constant obtained from applying the models were summarized in Tables 2 and 3.

  • Fitting of all adsorption model should be added.

Response: All adsorption models used in kinetics and isotherm studied were added, and their costant were illustrated in Tables 2 and 3.

  • How can you explain the sustainable removal of pollutant by adsorption. In general, people say that adsorption s just moving the pollutant from solution to the solid phase resulting in creation of secondary pollutant.

Response: Indeed, the natural materials such as cellulose and chitosan have replaced petroleum-derived polymers, offering natural and sustainable alternatives. Furthermore, these materials can be reused in the absorption after acid treatment to remove the adsorbed ions and make concentrated solutions from these ions. In addition, these materials are biodegradable, which can help in the degradation of the organic materials adsorbed on it.

Reviewer 2 Report

After reading the manuscript entitled: Preparation of isopropyl acrylamide grafted chitosan and carbon bionanocomposites for adsorption of lead ion and methylene blue, I regret to say that it does not meet the level of papers to be accepted in polymers due to the following reasons:

1- The Introduction is weak and there is no linking between the paragraphs and does not show the importance of what has been done in the report.

2- English editing is required.

3- It is more accurate to use the word adsorption rather than sorption or absorption, and adsorbent rather than sorbent.

4- Table 1 would be more useful if used to summarize the results of applying the models rather than showing information already available in the literature.

5- Figure 1 shows the SEM, TEM, and XRD for an only activated carbon samples, while it is important to compare the three prepared samples.

6- The quality of Figures 4-6 must be improved.

7- The discussion of the results is very shallow.

8- The conclusion should be rewritten to focus on the main findings and why they are important.

Author Response

Reviewer # 2:

Comments and Suggestions for Authors

After reading the manuscript entitled: Preparation of isopropyl acrylamide grafted chitosan and carbon bionanocomposites for adsorption of lead ion and methylene blue, I regret to say that it does not meet the level of papers to be accepted in polymers due to the following reasons:

  • The Introduction is weak and there is no linking between the paragraphs and does not show the importance of what has been done in the report.

Response: The introduction section was modified and rearranged.

  • English editing is required.

Response: The English was editing.

  • It is more accurate to use the word adsorption rather than sorption or absorption, and adsorbent rather than sorbent.

Response: The words sorption, absorption, and sorbent were corrected in the text.

  • Table 1 would be more useful if used to summarize the results of applying the models rather than showing information already available in the literature.

Response: Table 1 was deleted, and the constant obtained from applying the models were summarized in Tables 2 and 3.

  • Figure 1 shows the SEM, TEM, and XRD for an only activated carbon samples, while it is important to compare the three prepared samples.

Response: SEM and XRD of crosslinked chitosan and nanocomposite were performed and introduced in the paper.

  • The quality of Figures 4-6 must be improved.

Response: The quality of Figures was improved.

  • The discussion of the results is very shallow.

Response: The discussion section was improved.

  • The conclusion should be rewritten to focus on the main findings and why they are important.

Response: The conclusion part was fully revised.

Round 2

Reviewer 1 Report

accept

Reviewer 2 Report

The authors have done all the required revisions. I recommend the publication in its current shape.